# Small Foreign Object Debris Detection for Millimeter-Wave Radar Based on Power Spectrum Features

**DOI:** 10.3390/s20082316

**Published:** 2020-04-18

**Authors:** Peishuang Ni, Chen Miao, Hui Tang, Mengjie Jiang, Wen Wu

**Affiliations:** Ministerial Key Laboratory of JGMT, Nanjing University of Science and Technology, Xiao Ling Wei200#, Nanjing 210094, China

**Keywords:** FOD detection, feature extraction, millimeter-wave radar, the PSO algorithm, SVDD classifier

## Abstract

Foreign object debris (FOD) detection can be considered a kind of classification that distinguishes the measured signal as either containing FOD targets or only corresponding to ground clutter. In this paper, we propose a support vector domain description (SVDD) classifier with the particle swarm optimization (PSO) algorithm for FOD detection. The echo features of FOD and ground clutter received by the millimeter-wave radar are first extracted in the power spectrum domain as input eigenvectors of the classifier, followed with the parameters optimized by the PSO algorithm, and lastly, a PSO-SVDD classifier is established. However, since only ground clutter samples are utilized to train the SVDD classifier, overfitting inevitably occurs. Thus, a small number of samples with FOD are added in the training stage to further construct a PSO-NSVDD (NSVDD: SVDD with negative examples) classifier to achieve better classification performance. Experimental results based on measured data showed that the proposed methods could not only achieve a good detection performance but also significantly reduce the false alarm rate.

## 1. Introduction

Foreign object debris (FOD) refers to any object located in the airport environment that can injure or harm the aircraft. Typical examples of FOD include aircraft parts, twisted metal strips, ramp garbage, plastic products, and so on [1]. FOD poses a safety risk to aircraft and a significant economic loss to airlines. The estimated value of FOD damage reaches several billion USD per year for airlines in terms of structural damage and maintenance costs [2,3]. After the accident of a Concorde airplane due to a metal strip on the runway, FOD detection has become an exciting research area and gained significant attention [4].

At present, the outstanding FOD detection systems are the Tarsier system by QinetiQ, FODFinder system by Trex, iFerret system by Stratech, and FODetect system by Xsight [5,6,7,8]. These systems mainly realize FOD detection through radar or electro-optical technologies [9]. They follow the standards of an advisory circular issued by the Federal Aviation Administration (FAA), and the performance of these systems has been tested. For example, as a monitoring system relying only on video image processing, the iFerret system can detect targets with a height of 4 cm, and a target with a height of 2 cm can be detected under good weather conditions. However, this system is highly vulnerable to haze weather [5]. The FODFinder system adopts millimeter-wave radar and video image processing to detect FOD targets, but it can only detect targets as small as 2.5 cm × 2.5 cm (diameter × height) [7]. Thus, these systems only satisfy the demands of civil airports. For military airports, it is highly desirable to detect much smaller FOD targets, such as a 2 cm diameter metal ball, though this is extremely difficult. For FOD detection systems that rely on millimeter-wave radar, they are generally able to detect targets with a radar cross-section (RCS) of not less than −30 dBsm. However, for a 2 cm diameter metal ball, the RCS is as low as −35 dBsm, which is very difficult to detect. As the FOD target becomes smaller, its RCS will decrease to a very small value, resulting in the target echo being close to the ground clutter. Therefore, the signal-to-clutter ratio (SCR) is greatly reduced, which makes the detection difficulty increase sharply. This paper mainly aims to solve the difficult problem of small FOD detection in military airports.

There are several studies have been published in recent years, which mainly focus on image processing and radar signal processing. Different from the traditional image processing methods, novel algorithms based on a convolutional neural network (CNN) are proposed in References [10,11] to detect FOD based on optical images, which achieve a high recall rate for small FOD, such as small screws. However, the images obtained at night or in hazy weather are not ideal due to the influence of light and brightness. Yigit et al. detected a 1.2 cm × 3.5 cm (diameter × height) screw at a distance of 2 m from the radar by using millimeter-wave, ground-based synthetic aperture radar imaging, while the detection capability of remote small FODs has not been verified [12]. In radar signal processing methods, the constant false alarm rate (CFAR) detection algorithm is commonly utilized for FOD detection. Cell average (CA) and ordered statistics (OS) CFAR methods have good detection performance with a homogeneous background, while the detection effect for small FOD targets is reduced with non-homogeneous backgrounds [13,14]. The clutter map (CM) CFAR algorithm can estimate the average clutter power through multiple iterations but the “self-shielding” is inevitable, leading to a reduction in detection performance [15]. Leonardi et al. proposed the fusion of raw signal level data for the echo data received by multiple radars, which improved the detection rate of small FOD targets [16]. However, for a small target whose SCR is equal to 4 dB, the detection rate of the method only reaches 35%, which is still a very low value. Therefore, the principle of radar automatic target recognition was applied in this study to transform the target detection into classification, and then implement FOD detection through a classifier [17]. To verify the practical effect of this method, we collected data at a military airport in Jiangsu, China. Based on the measured data, we compared the methods proposed in this paper with the traditional CA-CFAR and CM-CFAR algorithms. For a 2 cm diameter metal ball, the results showed that the detection rate of the CM-CFAR algorithm was slightly higher than that of the CA-CFAR algorithm, but the false alarm rate of the CA-CFAR algorithm was lower. However, the methods proposed in this paper not only achieved a higher detection rate than the CM-CFAR algorithm, but also obtained a lower false alarm rate than the CA-CFAR algorithm. Therefore, it is helpful to realize the difficulty of detecting small FOD targets in military airfields.

This paper is arranged as follows: Section 2 introduces the signal model of the linear frequency modulated continuous wave (LFMCW) radar used in this study and outlines the principle of the traditional CFAR algorithms for FOD detection. In Section 3, the principle of the classifier is introduced, and it is applied for FOD detection. The feature extraction is carried out on the data first, then the theory of the feature-based SVDD classifier is expounded. Finally, we describe the process of optimizing the classifier parameters using the particle swarm optimization (PSO) algorithm. Section 4 gives the detection results and analysis of comparing traditional FOD detection with classifier-based FOD detection. Some conclusions are collected in Section 5.

Notations: In this paper, the bold uppercase letters are matrices, the bold lowercase letters are vectors, and the italic letters show scalar variables.

## 2. Traditional CA-CFAR for FOD Detection

### 2.1. Signal Model of LFMCW Radar

The typical LFMCW scheme is used for millimeter-wave FOD detection radar systems. The application of LFMCW has resulted in reduced costs, power consumption, and hardware and software complexity. For these reasons, LFMCW radar is attractive in near-field and high-resolution applications. For a frequency sweep cycle, the saw-tooth LFMCW signal emitted by the radar can be expressed as:(1)$$s_{t}(t)=A\exp\left[j2\pi \left(f_{0}t+\frac{1}{2}\mu t^{2}\right)\right],$$
where A and $f_{0}$ are the amplitude and center frequency of the transmission signal, respectively; $\mu =B_{r}/T_{r}$ is the sweeping slope; $B_{r}$ is the bandwidth of the sweep; and $T_{r}$ is the duration of a single frequency sweep. For a stationary FOD target on the airport runway, the echo signal received by the radar after the target reflection can be written as:(2)$$s_{r}\left(t\right)=K_{a}A\exp\left\{j2\pi \left[f_{0}\left(t-\tau \right)+\frac{1}{2}\mu \left(t-\tau^{2}\right)\right]\right\},$$
where $K_{a}$ is the reflection coefficient and τ=2R/c  is the two-way time delay for a FOD target located at range R. Substituting τ into Equation (2), the echo signal is down-converted with the transmitted signal, then the beat frequency signal can be obtained as follows:(3)$$s_{b}\left(t\right)=\frac{K_{a}A^{2}}{2}\exp\left[j2\pi \left(\mu \frac{2R}{c}t-\mu \frac{2R^{2}}{c^{2}}+\frac{2Rf_{0}}{c}\right)\right].$$

It can be seen that the signal $s_{b}\left(t\right)$ is a single-frequency signal with the frequency f=2μR/c. Then, performing an N-point fast Fourier transform (FFT) on it, and the sampling frequency is $f_{s}$. The signal after the FFT can be expressed as:(4)$$\begin{array}{l} S_{b}\left(n\right)=\\ \frac{K_{a}A^{2}N}{2}\sin c\left(n-\frac{2\mu R}{c}\frac{N}{f_{s}}\right)\exp\left\{-j\left[\frac{N-1}{N}\left(n-\frac{2\mu R}{cf_{s}}\frac{N}{N-1}\right)\pi \right]\right\}\exp\left[j2\pi \left(\frac{2Rf_{0}}{c}-\mu \frac{2R^{2}}{c^{2}}\right)\right] \end{array}$$
where sinc(x)=sin(πx)/(πx) and n=1, 2, ⋯, N−1. After the FFT processing, the range of the FOD target can be estimated from the spectrum of the $S_{b}\left(n\right)$ signal.

### 2.2. Traditional CFAR Algorithm

The constant false alarm rate (CFAR) algorithm is a typical method for FOD detection. Consider a CA-CFAR detector, the basic principle of which is shown in Figure 1 [18], where $s_{b}\left(t\right)$ is the beat frequency signal and $S_{b}\left(n\right)$ is the signal after the FFT, which is fed into the CA-CFAR detector. The shadow parts on both sides of the cell under test (CUT) are guard cells, which do not participate in the estimation of the ground clutter power to avoid missing detection. The detection threshold can be expressed as shown in Equation (5) where $p_{i},\;q_{i}\left(i,\;j=1,\;\cdots ,\;n\right)$ are the reference cells on both sides of the CUT; the power summation of the front and back edge reference cells are P and Q, respectively; Z is the average value of all the reference cells; and γ is the threshold coefficient:(5)$$T=\gamma Z=\frac{1}{2n}\gamma \left(P+Q\right)=\frac{1}{2n}\gamma \left(\sum_{i=1}^{n}p_{i}+\sum_{j=1}^{n}q_{j}\right).$$

The CA-CFAR detector outputs the detection result as either FOD present or FOD absent according to whether the power of the CUT is greater than the threshold. For the CM-CFAR detector, it mainly includes two steps: updating the clutter map and detecting the target [18]. First, the clutter map is obtained through multiple iterations of ground clutter, and then the detection threshold is determined. Finally, similar to the CA-CFAR detector, the power of the CUT is compared with the threshold to obtain the detection result.

## 3. FOD Detection with the PSO-SVDD Classifier

FOD detection involves binary hypothesis testing and can also be regarded as a binary classification [19]. There are only two different decision results, namely FOD present or FOD absent. The small FOD targets detection model can be simply represented as:(6)$$\left\{ \begin{array}{l} \mathrm{FOD}\;\mathrm{absent}:\;\;\;S_{b}\left(n\right)=S_{c}\left(n\right)\;\;\\ \mathrm{FOD}\;\mathrm{present}:\;\;S_{b}\left(n\right)=S_{c}\left(n\right)+S_{F}\left(n\right) \end{array}\right.$$
where $S_{c}\left(n\right)$ and $S_{F}\left(n\right)$ are the signals of ground clutter and the FOD target, respectively, and they are independent of each other. Therefore, this study aimed to use the PSO-SVDD classifier to replace the CFAR detectors for FOD detection. The data of signal $S_{b}\left(n\right)$ was categorized into two kinds, one was the data corresponding to ground clutter (with FOD absent), and the other was the data that contained the FOD target (with FOD present). As shown in Figure 2, the FOD absent data were taken as the training data in the training stage. First, the power spectrum features of each training sample were extracted and combined as an eigenvector to describe it. Then, all the eigenvectors were combined into an eigenvector matrix to train the SVDD classifier. Meanwhile, the PSO algorithm was used to optimize the parameters of the classifier. In the testing stage, the FOD absent and FOD present data were utilized to validate the classification performance. For each test sample, the power spectrum features were also extracted, and the eigenvector was input into the trained PSO-SVDD classifier to obtain the final classification result: FOD present or FOD absent.

### 3.1. Feature Extraction

As shown in Figure 3, for the cell $S_{b}\left(u\right)$ under test, k cells before and after it were combined into a small block of data, and feature extraction was carried out on the block of data.

The data in this block can be written as $S=\left[S_{b}\left(u-k\right),\cdots ,S_{b}\left(u-1\right),S_{b}\left(u\right),S_{b}\left(u+1\right),\cdots ,S_{b}\left(u+k\right)\right]$, mapping it into the power spectrum domain; then, the power spectrum of it can be expressed as $D=S^{2}=\left[D\left(u-k\right),\cdots ,D\left(u-1\right),D\left(u\right),D\left(u+1\right),\cdots ,D\left(u+k\right)\right]$. The power spectrum features can be extracted using the following equation:(7)$$f_{c}=\sum_{m=u-k}^{u+k}{\left(m-\tilde{m}\right)}^{2}v\left(m\right),$$
which represents the second-order central moment of the power spectrum, where $\tilde{m}=\sum_{m=u-k}^{u+k}mv\left(m\right)$ denotes the first-order central moment of the power spectrum and $v\left(m\right)=D\left(m\right)/\sum_{m=u-k}^{u+k}D\left(m\right)$ is the normalized power spectrum.
(8)$$f_{a}=\frac{1}{2k+1}\sum_{m=u-k}^{u+k}D\left(m\right)$$
represents the average power spectrum. Dimensionality reduction was achieved by extracting the two power spectrum features from the data, and the data with FOD present and FOD absent was converted into the power spectrum domain with higher discrimination. The two features could be combined into an eigenvector of $x=\left(f_{c},\;f_{a}\right)$ to describe the block of data.

### 3.2. Principle of SVDD

SVDD is a one-class classifier proposed by Tax and Duin, which is widely used in anomaly detection [20,21]. The SVDD models a class of data by fitting a hypersphere with center a and radius r around all or most of the samples. Assume that we are given a set of training samples $\left\{x_{i},\;i=1,\;2,\;\cdots ,\;l\right\}$, where l is the number of samples. The SVDD aims to minimize the volume of the hypersphere by minimizing $r^{2}$. However, the training data will not be spherical in many cases; therefore, the kernel function is introduced to map it into the kernel space. The kernel function can be written as:(9)$$K\left(x_{i},x_{j}\right)=\langle \Phi \left(x_{i}\right),\Phi \left(x_{j}\right)\rangle =\Phi {\left(x_{i}\right)}^{T}\Phi \left(x_{j}\right),$$
where $\left\{\Phi \left(x_{i}\right),\;i=1,\;2,\;\cdots ,\;l\right\}$ is the data set mapped into the kernel space. The solution of the optimal decision surface of SVDD can be expressed as the following optimization problem:(10)$$\left\{ \begin{array}{l} \;\underset{r,a,\xi_{i}}{\min}\;\;\;F\left(r,a\right)=r^{2}+C\sum_{i=1}^{l}\xi_{i}\\ \;s.t.\;\;\;\;{\left\|\Phi \left(x_{i}\right)-a\right\|}^{2}\leq r^{2}+\xi_{i},\;\xi_{i}\geq 0,\;\forall i=1,\;2,\;\cdots ,\;l \end{array}\right..$$

The C parameter controls the tradeoff between the volume of the hypersphere and the number of target objects rejected. Since the training data may contain outliers, $\xi_{i}$ are slack variables that relax the constraints. A lower classification error rate and a smoother decision boundary containing the training samples can be obtained by adjusting C. The constraints can be incorporated into the objective function by using Lagrange multipliers:(11)$$L\left(r,a,\xi_{i},\alpha_{i},\beta_{i}\right)=r^{2}+C\sum_{i=1}^{l}\xi_{i}-\sum_{i=1}^{l}\beta_{i}\xi_{i}-\sum_{i=1}^{l}\alpha_{i}\left\{r^{2}+\xi_{i}-\left\|\Phi \left(x_{i}\right)-a\right\|\right\},$$
where $\alpha_{i}$ and $\beta_{i}$ are Lagrange multipliers, where $\alpha_{i}$, $\beta_{i}\geq 0$. The above optimization is a minimax problem, which can be written as:(12)$$\underset{\alpha_{i},\beta_{i}}{\max}\;\underset{r,a,\xi_{i}}{\min}\;L\left(r,a,\xi_{i},\alpha_{i},\beta_{i}\right).$$

L should be minimized with respect to r, a, and $\xi_{i}$ and maximized with respect to $\alpha_{i}$ and $\beta_{i}$. Setting the partial derivatives to zero gives the constraints:(13)$$\left\{ \begin{array}{l} \;\frac{\partial L}{\partial r}=0:\;\;\;\;\sum_{i=1}^{l}\alpha_{i}=1\\ \;\frac{\partial L}{\partial a}=0:\;\;\;\;a=\sum_{i=1}^{l}\alpha_{i}x_{i}\\ \;\frac{\partial L}{\partial \xi_{i}}=0:\;\;\;\;C-\alpha_{i}-\beta_{i}=0 \end{array}\right..$$

Substituting Equation (13) into Equation (11) and introducing the kernel function, the final optimization problem can be obtained as follows:(14)$$\left\{ \begin{array}{l} \;\underset{\alpha }{\max}\;L=\sum_{i=1}^{l}\alpha_{i}K\left(x_{i},x_{i}\right)-\sum_{i,j=1}^{l}\alpha_{i}\alpha_{j}K\left(x_{i},x_{j}\right)\\ \;s.t.\;\;\sum_{i=1}^{l}\alpha_{i}=1,\;0\leq \alpha_{i}\leq C,\;i=1,\;2,\;\cdots ,\;l \end{array}\right..$$

According to the Karush–Kuhn–Tucher (KKT) conditions, only part of $\alpha_{i}$ satisfies $0<\alpha_{i}\leq C$ and its corresponding $x_{i}$ is the support vector. The radius and center of the hypersphere are both determined by the support vectors. Suppose the number of support vectors is nSV, and SVs denotes the set of support vectors; then, the radius and center of the hypersphere are:(15)$$\left\{ \begin{array}{l} \;r^{2}=K\left(x_{i},x_{i}\right)-2\sum_{i=1}^{nSV}\alpha_{i}K\left(x_{i},x_{i}\right)+\sum_{i,j=1}^{nSV}\alpha_{i}\alpha_{j}K\left(x_{i},x_{j}\right)\\ \;a=\sum_{i=1}^{l}\alpha_{i}\Phi \left(x_{i}\right) \end{array}\right..\;$$

In the SVDD model, to determine whether a new sample $x^{*}$ lies in the description, the distance from the center of the sphere to $x^{*}$ must be less than $r^{2}$. Hence, the decision function for $x^{*}$ is: (16)$$d\left(x^{*}\right)={\left\|\Phi \left(x^{*}\right)-\Phi \left(a\right)\right\|}^{2}-r^{2}.$$

Consequently, the discriminant result for the FOD detection is: (17)$$\left\{ \begin{array}{l} \;d\left(x^{*}\right)\leq 0,\;\;\;\;\mathrm{FOD}\;\mathrm{absent}\\ \;d\left(x^{*}\right)>0,\;\;\;\;\mathrm{FOD}\;\mathrm{present} \end{array}\right..$$

Several different choices of kernel functions exist. In this study, we used the well-known Gaussian kernel function. The Gaussian kernel has only one free parameter to be tuned and is shown to yield tighter boundaries than other kernel choices [22]. The Gaussian kernel is given by the following:(18)$$K\left(x,y\right)=\exp\left(-{\left\|x-y\right\|}^{2}/2\sigma^{2}\right),$$
where σ is a free parameter that is adjusted to control the tightness of the boundary and is typically optimized through cross-validation [23]. In the kernel-based SVDD, the selection of free parameters C and σ affects the classification performance; therefore, they need to be optimized to achieve the best detection performance.

### 3.3. Parameter Optimization Based on the PSO

The grid method is adopted in traditional SVDD parameter optimization, which is time-consuming and may not find the optimal parameters. This study applied the PSO algorithm for parameter optimization, which is a global search evolutionary algorithm with a simple structure and low complexity [24]. The purpose of the PSO algorithm is to obtain the optimal parameters C and σ to minimize the classification error rate. For a testing set with N samples, its classification error rate is:(19)$$err=\frac{FP+FN}{N},$$
where FP is the number of samples where FOD is judged as FOD absent and FN is the number of samples where FOD is judged as FOD present. The steps of the PSO can be described in Algorithm 1.
**Algorithm 1.** The optimization steps of parameters using PSO.**Step 1:** Input the training and testing data, initialize the parameters C and σ of SVDD model, and set the searchable range of the parameters.**Step 2:** Initialize the particle swarm, including the population size W, acceleration constants $c_{1}$ and $c_{2}$, inertia weight ω, maximum number of iterations $I_{t}$, and the particle speed and position.**Step 3:** Determine the individual extremum of the initial position and the optimal position of the particle swarm. **Step 4:** Calculate the fitness value of the new position of each particle in the swarm.**Step 5:** Compare the current optimal position of each particle with the optimal position of the particle swarm and update the optimal solution to the current optimal position of particle swarm.**Step 6:** Update the speed and position of the particle.**Step 7:** Determine whether the SVDD model with the current parameters can minimize the error rate or reach the maximum number of iterations. If one of them is satisfied, the optimal parameters C and σ are obtained. Otherwise, return to step 4 to recalculate the particle fitness value.

## 4. Detection Results and Analysis

In this section, the simulations that were conducted to compare the PSO-SVDD classification method with the classical CA-CFAR and CM-CFAR algorithms are described. To obtain the measured data, a millimeter-wave radar was used to collect data at a military airport in Jiangsu, China, where the airport runways were paved with asphalt. The operation and the signal processing parameters of the radar system were as follows: the operation frequency was $f_{0}=94.5\;\mathrm{GHz}$, the bandwidth of the sweep was $B_{r}=600\;\mathrm{MHz}$, the duration of a single frequency sweep was $T_{r}=1\;\mathrm{ms}$, the transmitting power was 20 dBm, and the vertical and horizontal beamwidth of the antenna was 8° and 2°, respectively. The sampling frequency was $f_{s}=1\;\mathrm{MHz}$ and the number of points for the FFT was N=1024. The radar sensor was placed at 1.5 m above the ground. Additionally, during the feature extraction, the number of the intercepted cells was set to k=6. For the CA-CFAR detector, there were two guard cells and four reference cells on either side. According to the principle of CA-CFAR, the threshold coefficient was set to γ=3.5 in this paper.

Before collecting the data, the experimental scene was manually checked first to ensure that there were no FOD targets. Then, the millimeter-wave radar scanned the experimental scene and recorded the echo data of the ground clutter. After that, FOD targets were manually placed on the airport runway. Taking the detection of a 2 cm diameter metal ball as an example, we placed the target at a distance of 42 m from the radar. Then, the echoes of it were acquired using radar scanning. Figure 4a shows the amplitude spectrum when the metal ball was present and Figure 4b depicts the amplitude spectrum of the ground clutter (without the metal ball). It can be seen from Figure 4 that in the area closer to the radar, since the energy of the echo entered the radar from the sidelobe, the energy of the ground clutter echo was relatively strong. However, in the area far from the radar, the target echo received by the radar was weaker, which further increased the difficulty of detecting small FOD targets. According to the local magnified image near 42 m in Figure 4a, the SCR of the target was only 4 dB, which was almost submerged in the ground clutter. Since the ground clutter echo and the target echo were difficult to distinguish, it was very difficult to detect the target via the CA-CFAR and CM-CFAR algorithms. Therefore, we extracted the implicit internal features of the ground clutter echo and target echo to further enlarge the distinction between them, and then the classification method could be applied for the FOD detection.

Based on the method proposed in this paper, the two power spectrum features were extracted. Figure 5a,b shows the distribution of the two features when the FOD was present and absent. The second-order central moment describes the distribution of the power spectrum energy relative to its geometric centroid. The more dispersed the energy distribution, the smaller the second-order central moment. From Figure 5a, when the FOD was present, the second-order central moment was smaller than that when it was absent. For the average power spectrum, the greater the energy, the greater the average. From Figure 5b, it was obvious that the average power spectrum in the presence of the FOD was greater than that in the absence of FOD. Compared with Figure 4, by extracting these two features, the discriminability between FOD present and FOD absent was significantly improved. Then, the extracted features could be used to train the classifier or test the performance of it.

To verify the FOD detection performance of the PSO-SVDD detector, we counted 300 samples of ground clutter data and 150 samples of data containing FOD. In the training stage, 80% of the FOD absent samples were randomly selected to train the PSO-SVDD classifier, and the rest were taken as the testing data. Based on the analysis in Section 3.3, the parameters were first set to W=30, $c_{1}=2$, $c_{2}=2$, ω=0.5, $I_{t}=200$, 0<C<1, and $0<\sigma <2^{8}$. Then, the PSO algorithm was used to optimize the parameters of the SVDD model. The optimization process is shown in Figure 6, where the minimum classification error rate was 25.71% after 11 iterations, and the optimal decision boundary could be obtained with parameters C=0.63 and σ=63.77. In the classification process, the test samples inside the decision boundary were judged FOD absent; otherwise, they were judged as FOD present.

The classification results of the PSO-SVDD are depicted in Figure 7. In Figure 7a, the red circle markers denote the feature vectors of the samples without FOD, which were utilized to train the PSO-SVDD classifier. In Figure 7a,b, the green circle markers represent the support vectors, and the decision boundary is denoted by the purple line. The decision boundary distinguished the FOD present samples from FOD absent samples in the testing procedure. It can be seen from Figure 7b that there were two FOD absent samples that were classified outside the decision boundary, that is, they were judged as FOD present, resulting in false alarms. Furthermore, the remaining samples with FOD absent were all divided into the interior of the decision boundary, indicating that the judgment was correct. Meanwhile, most of the FOD present samples were correctly divided outside the decision boundary, while a small part of them were wrongly classified inside as FOD absent.

Since only the FOD absent samples were utilized to train the classifier in the training procedure, overfitting inevitably occurred. To improve the performance further, 10% of the samples containing FOD were randomly added to train a PSO-NSVDD (NSVDD: SVDD with negative examples) classifier. The principle of the NSVDD model is similar to that of the SVDD model [25], while the parameters that need to be optimized are $C_{+}$, $C_{-}$, and $\sigma^{\prime }$, where $C_{+}$ and $C_{-}$ are the penalty parameters for the FOD absent and FOD present samples, respectively, and $\sigma^{\prime }$ is the kernel width of the NSVDD model. After 23 iterations, the optimal parameters of the model were $C_{+}=\;0.76$, $C_{-}=0.58$, and $\sigma^{\prime }=12.87$. The final detection results of the PSO-NSVDD classifier are shown in Figure 8, where the yellow circles in Figure 8a indicate the newly added FOD present samples. Furthermore, it can be seen from Figure 8b that, due to the addition of samples with FOD present in the training stage, the decision boundary was described more accurately and the overfitting was avoided. Therefore, compared with the results of the PSO-SVDD detector in Figure 7b, more samples with FOD present were classified outside the decision boundary and the FOD detection performance was further improved. Meanwhile, almost all the FOD absent samples were correctly classified within the decision boundary, which further reduced the false alarm rate.

To detect a target that exists at any distance from the radar, according to the method proposed in this paper, we first divided the collected ground clutter data into several small blocks in the range dimension. Similarly, we also divided the data containing the minimum FOD target to be detected into small blocks. Then, the feature extraction was performed on each block, and finally, the optimal PSO-SVDD and PSO-NSVDD classifiers were trained in these blocks. Within each block, there was little variation of ground clutter echo and the PSO-NSVDD classifier trained by the data containing the minimum FOD target could also be used to detect larger targets. Therefore, the parameters of these classifiers could be fixed. In the later detection process, there was no need to retrain new classifiers, which meant that these trained classifiers could be directly utilized to detect the FOD targets that existed at any distance. Therefore, we placed a 2 cm diameter metal ball and a 4.3 cm diameter golf ball at a distance of 10 m from the radar, and placed another golf ball with the same size at a distance of 42 m to verify the detection effect. We also counted 150 samples containing these FOD targets for testing. Table 1 shows the detection results of these targets using four detection methods. For the metal ball and golf ball, SCR = 19 dB and 30 dB at 10m and SCR = 4 dB and 13 dB at 42 m, respectively. According to the detection rate $P_{d}$ and the false alarm rate $P_{f}$ of the metal ball at different ranges, compared with the CA-CFAR algorithm, the detection rate of the CM-CFAR algorithm was slightly higher, but the false alarm rate was also higher, which was not what we expected. For the CA-CFAR and CM-CFAR algorithms, the golf ball was detected much easier than the metal ball, suggesting that it was difficult for the conventional CFAR methods to detect such small-volume FOD targets. However, for the metal ball at two ranges, the $P_{d}$ of the PSO-SVDD classifier was significantly higher, reaching 87.33% at 10 m and 64.67% at 42 m, and the $P_{f}$ was significantly lower for both. Additionally, after adding the samples with FOD present, the $P_{d}$ of the PSO-NSVDD was further increased to 92.61% at 10 m and 78.18% at 42 m. Meanwhile, for both the golf ball and metal ball, the PSO-NSVDD classifier produced a much lower false alarm rate. Based on the analysis above, we have reason to believe that the methods proposed in this paper not only showed a higher detection rate, but also a lower false alarm rate for the target with the low SCR. For the target with the high SCR, the detection rate was the same and the false alarm rate was even lower. Thus, these proposed methods were effective for a wide SCR range.

Finally, the two methods proposed in this paper were comprehensively analyzed as follows. For the PSO-SVDD classifier, only the ground clutter (with FOD absent) data needed to be utilized in the training stage. In an actual airport environment, a mass of ground clutter data can be collected to train the classifier; therefore, this method is easily implemented. For the PSO-NSVDD classifier, a small number of samples with FOD present needed to be added during the training stage, meaning that the data containing the minimum FOD target needed to be collected in advance, while it is unrealistic to collect such data in practical applications. However, the detection performance of the latter was further improved. Furthermore, compared with the traditional CA-CFAR and CM-CFAR methods, the detection performance of these two methods for small FOD targets was much higher.

## 5. Conclusions

Small FOD target detection under strong ground clutter is a key issue for millimeter-wave radar FOD detection, which is difficult to achieve using traditional CFAR algorithms. This paper transformed FOD detection into a kind of classification, and the PSO-SVDD and PSO-NSVDD classifiers were used to classify the FOD target and ground clutter to detect FOD from the background. The experimental results based on the measured data showed that the proposed methods achieved a higher detection rate and lower false alarm rate for small FOD targets than the classical CFAR algorithms.

## Figures and Tables

**Figure 1 sensors-20-02316-f001:**
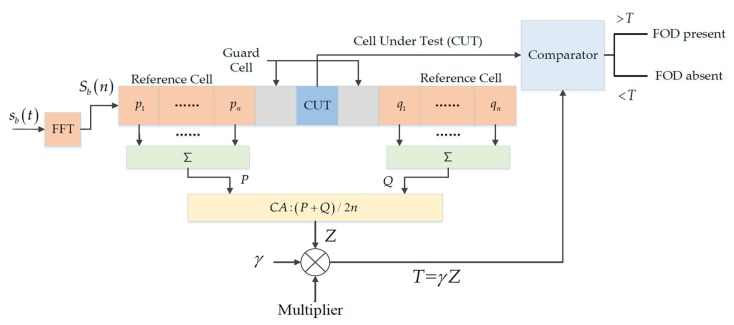
Basic principle of the cell average constant false alarm rate (CA-CFAR) detector. FFT: Fast Fourier Transform, FOD: Foreign Object Debris.

**Figure 2 sensors-20-02316-f002:**
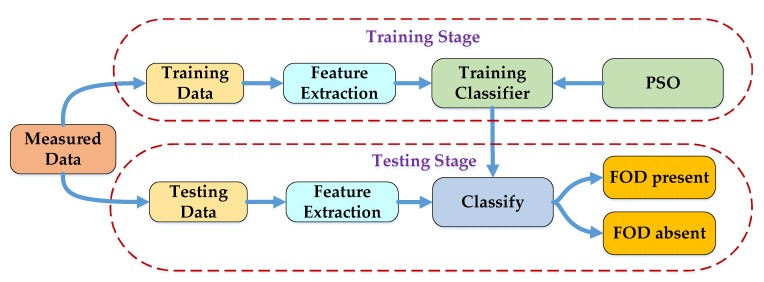
Flowchart of the FOD classification. PSO: Particle Swarm Optimization.

**Figure 3 sensors-20-02316-f003:**
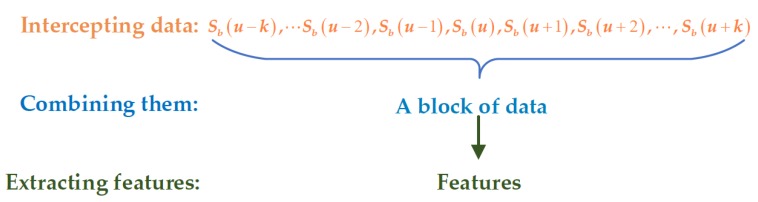
Flowchart of the feature extraction.

**Figure 4 sensors-20-02316-f004:**
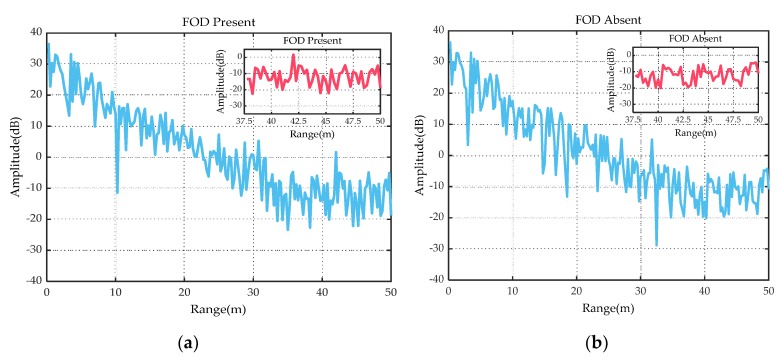
(**a**) Amplitude spectrum with FOD present. **(b**) Amplitude spectrum with FOD absent.

**Figure 5 sensors-20-02316-f005:**
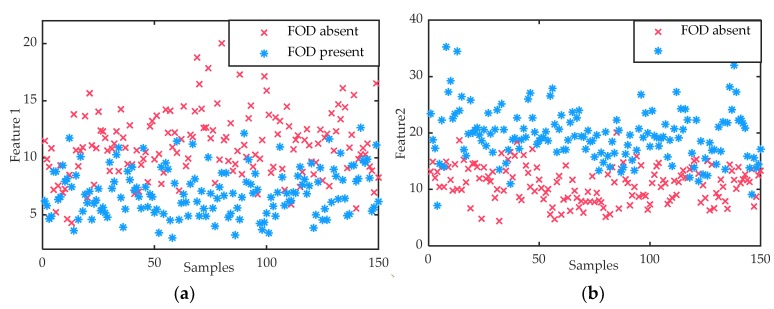
(**a**) Feature 1: the second-order central moment of the power spectrum. (**b**) Feature 2: the average power spectrum.

**Figure 6 sensors-20-02316-f006:**
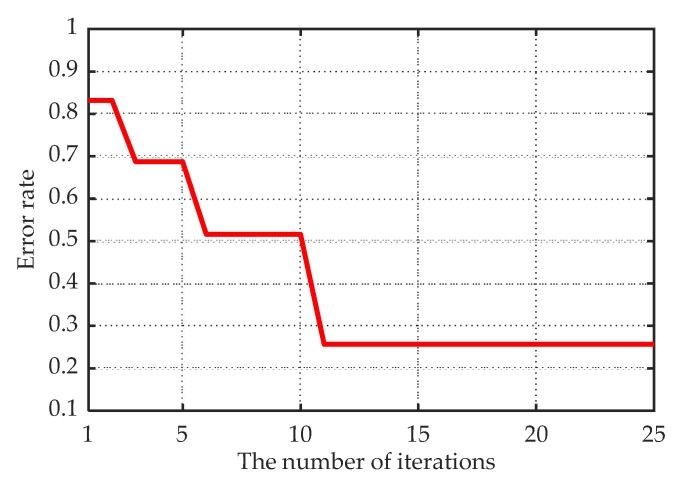
The optimization process of the PSO algorithm.

**Figure 7 sensors-20-02316-f007:**
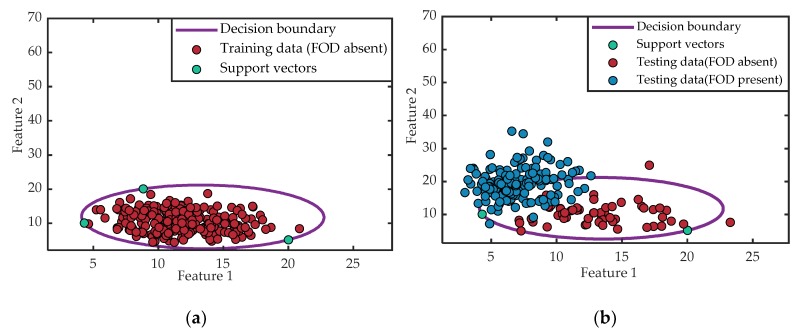
Detection results with the PSO support vector domain description (PSO-SVDD): (**a**) training procedure and (**b**) testing procedure.

**Figure 8 sensors-20-02316-f008:**
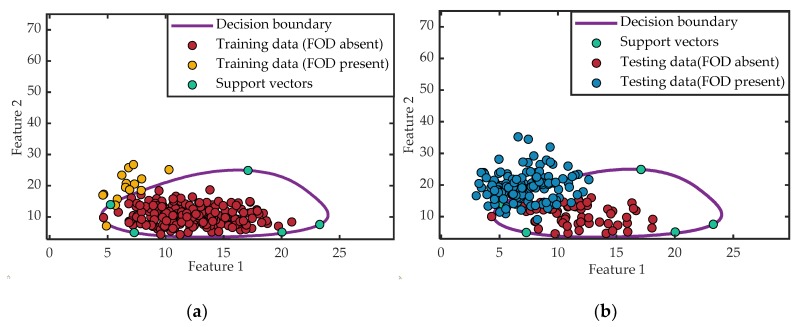
Detection results with the PSO-NSVDD: (**a**) training procedure and (**b**) testing procedure.

**Table 1 sensors-20-02316-t001:** The detection results using different methods.

Methods	Metal Ball d=2 cm,R=10 m		Golf Ball d=4.3 cm,R=10 m		Metal Ball d=2 cm,R=42 m		Golf Ball d=4.3 cm,R=42 m	
	$P_{d}\left(\%\right)$	$P_{f}\left(\%\right)$	$P_{d}\left(\%\right)$	$P_{f}\left(\%\right)$	$P_{d}\left(\%\right)$	$P_{f}\left(\%\right)$	$P_{d}\left(\%\right)$	$P_{f}\left(\%\right)$
CA-CFAR	46.67	3.2	100	1.35	13.33	4.67	100	1.11
CM-CFAR	53.14	7.41	100	4.58	21.57	7.68	100	3.14
PSO-SVDD	87.33	1.25	100	0.65	64.67	0.95	100	0.83
PSO-NSVDD	92.61	0.39	100	0.1	78.18	0.51	100	0.12

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
