# Peer review of "Small Foreign Object Debris Detection for Millimeter-Wave Radar Based on Power Spectrum Features"

_sensors, 2020, doi:10.3390/s20082316_

Round 1

Reviewer 1 Report

This paper includes interesting proposal of small FOD detection method based on millimeter-radar system. However, the additional descriptions are required for the publication.

In 1. Introduction, the authors stated that the 2 cm diameter metal ball is difficult to detect commercially available FOD detection system. The radar cross section of the 2 cm diameter metal ball is -29 dBsm in all directions. It is not too small RCS values for the commercially available FOD detection systems. In addition, the authors stated that the performance of the FOD detection system is not satisfy the requirement of military airports. Could you add comments on the detection performance of the radar system and the requirement of military airport?

In 3.1 Feature Extraction, the authors show the amplitude spectrum of the millimeter-wave radar. However, there is no descriptions on the operation and the signal processing parameters of the radar system. Could you add these parameters such as operation frequency, bandwidth, transmitting power, antenna gain, number of integrations, and range resolutions? In addition, the signal-to-noise ratio (SNR) of the detected targets are not clear in Figure 3. To start the discussion, please add the SNR of all detected targets throughout the manuscript. It is better to use unit in dB for the vertical axis of Figure 3.

In 4 Detection Results and Analysis, the authors show the improvements of proposed PSO-NSVDD method. However, as described in the previous sentence, the SNR of the samples are required to be mentioned. Please describe the proposed PSO-NSVDD method is effective for the wide SNR range or specific SNR range.

Reviewer 2 Report

(1)More detail characteristics of the CW radar utilized in the experiment is needed. (2)Surface clutter is potentially affected quite strongly by the surface materials and condition. The authors should specify the type of surface used in the experiments and comment on the likely effect of different surface types and conditions on the operation of this scheme. (3)In Figure 3, the amplitude of the clutter within the range of 15m is stronger than other ranges, how to deal with it? (4) The authors should give more description about the detection results of CA-CFAR and PSO-SVDD.

Reviewer 3 Report

I like to suggest the following:

1) improve the context desription, till page 3 I dont understand if you operate using a real radar or by simulations

2) add "Deb-Ra by Rehinmetall" as an actual FOD detection system using this reference:

Leonardi M., Fastella V., Piracci E.G.;W-Band Multi-Radar processing for Airport Foreign Object Debris and Humans Detection; EuMCE 2019

Reviewer 4 Report

The paper is an interesting one addressing the needs of proper identification of aerial objects. However the paper feels incomplete mainly because of lack of enough useful info on how to do the proper identification at any distance.  In particular, the paper demonstrates FOD at 42 metres. There's a significant work done in order to train the system to identify the object at 42 M given that the object itself is about 2 cm in diameter. There are few critical issues here.  First, it's not clear how system is trained in cases where the object is a bit smaller or a bit larger. No test or validation data is provided on such scenarios. Second, it's not clear how the system accounts for existing objects at closer distance or a larger distance. There is no test info provided on resulting identification for an object say at 20 metres rather than 42 metres given the training data set used here. Third, the paper doesn't address the addition of potential material impact on the training and thus the identification.  This is a scenario that is likely to happen in cases where there is for example rain or snow conditions. 

Overall the paper is interesting but lacks addressing multiple practical fundamental implementation issues that makes it less useful to the reader. I strongly encourage the writers to address these issues before resubmitting a revised version of this manuscript.

Round 2

Reviewer 1 Report

I would like to thank the authors for sufficiently addressing all of my concerns and comments.

Author Response

Your efforts in the review process of our manuscript are greatly appreciated. Thanks again for your good comments.  

Reviewer 2 Report

1) The detection results should be compared with Clutter Map CFAR.

2) The authors should give more detailed description about Fig. 4 , Fig. 5 and two Features. From Fig.4, we believe the FOD present or absent will not affect the value of the two features.  
